# Retrospective Analysis of the Detection of Pathogens Associated with the Porcine Respiratory Disease Complex in Routine Diagnostic Samples from Austrian Swine Stocks

**DOI:** 10.3390/vetsci10100601

**Published:** 2023-10-02

**Authors:** René Renzhammer, Angelika Auer, Igor Loncaric, Annabell Entenfellner, Katharina Dimmel, Karin Walk, Till Rümenapf, Joachim Spergser, Andrea Ladinig

**Affiliations:** 1Department for Farm Animals and Veterinary Public Health, University Clinic for Swine, University of Veterinary Medicine, Veterinärplatz 1, 1210 Vienna, Austria; rene.renzhammer@vetmeduni.ac.at; 2Institute of Virology, Department of Pathobiology, University of Veterinary Medicine, Veterinärplatz 1, 1210 Vienna, Austria; angelika.auer@vetmeduni.ac.at (A.A.); katharina.dimmel@vetmeduni.ac.at (K.D.); till.ruemenapf@vetmeduni.ac.at (T.R.); 3Institute of Microbiology, Department of Pathobiology, University of Veterinary Medicine, Veterinärplatz 1, 1210 Vienna, Austria; igor.loncaric@vetmeduni.ac.at (I.L.); joachim.spergser@vetmeduni.ac.at (J.S.); 4Tierklinik Sattledt, 4642 Sattledt, Austria

**Keywords:** PRDC, respiratory pathogens, pigs, detection rate, specimens, lung, oral fluids, coinfections, PRRSV, *M. hyopneumoniae*

## Abstract

**Simple Summary:**

Multiple viruses and bacteria can cause respiratory disease in pigs. We aimed to report how frequently certain viruses and bacteria were detected in samples from pigs with respiratory disease in the course of routine diagnostic procedures at the University of Veterinary Medicine in Vienna between 2016 and 2021. While *Mycoplasma* (*M.*) *hyorhinis* (55.1%) had the highest detection rate, influenza A virus had the lowest detection rate (6.1%) in the investigated samples. Lung samples tested positive for PRRSV RNA were also more likely to be positive for *M*. *hyopneumoniae* and *Pasteurella* (*P*.) *multocida*. Samples tested positive for *M. hyopneumoniae* were more likely to be positive for *P. multocida* and *Streptococcus suis*, but less likely to be positive for *M. hyorhinis*. In conclusion, lung samples that were positive for a primary pathogenic agent were more likely to be positive for a secondary pathogenic agent.

**Abstract:**

The diagnostic workup of respiratory disease in pigs is complex due to coinfections and non-infectious causes. The detection of pathogens associated with respiratory disease is a pivotal part of the diagnostic workup for respiratory disease. We aimed to report how frequently certain viruses and bacteria were detected in samples from pigs with respiratory symptoms in the course of routine diagnostic procedures. Altogether, 1975 routine diagnostic samples from pigs in Austrian swine stocks between 2016 and 2021 were analysed. PCR was performed to detect various pathogens, including porcine reproductive and respiratory syndrome virus (PRRSV) (n = 921), influenza A virus (n = 479), porcine circovirus type 2 (PCV2) (n = 518), *Mycoplasma* (*M*.) *hyopneumoniae* (n = 713), *Actinobacillus pleuropneumoniae* (n = 198), *Glaesserella* (*G.*) *parasuis* (n = 165) and *M. hyorhinis* (n = 180). *M. hyorhinis* (55.1%) had the highest detection rate, followed by PCV2 (38.0%) and *Streptococcus* (*S*.) *suis* (30.6%). PRRSV was detected most frequently in a pool of lung, tonsil and tracheobronchial lymph node (36.2%). *G. parasuis* was isolated more frequently from samples taken after euthanasia compared to field samples. PRRSV-positive samples were more likely to be positive for PCV2 (*p =* 0.001), *M. hyopneumoniae* (*p* = 0.032) and *Pasteurella multocida* (*p* < 0.001). *M. hyopneumoniae*-positive samples were more likely to be positive for *P. multocida* (*p* < 0.001) and *S. suis* (*p =* 0.046), but less likely for *M. hyorhinis* (*p* = 0.004). In conclusion, our data provide evidence that lung samples that were positive for a primary pathogenic agent were more likely to be positive for a secondary pathogenic agent.

## 1. Introduction

Respiratory disease in pigs still remains one of the most important health concerns in pig production, leading to high economic losses due to increased mortality rates and decreased growth rates [1]. The porcine respiratory disease complex (PRDC) of pigs is caused by complex interactions of infectious and non-infectious causatives [1,2]. Infectious agents include viruses like the influenza A virus (IAV), the porcine reproductive and respiratory syndrome virus (PRRSV), the porcine circovirus type 2 (PCV2), the pseudorabies virus (PRV), the classical swine fever virus (CSFV), and to a lesser extent, the porcine respiratory corona virus (PRCV), and the porcine cytomegalovirus (PMCV), as well as bacteria like *Mycoplasma* (*M*.) *hyopneumoniae* and *Actinobacillus* (*A.*) *pleuropneumoniae* [1,3]. Those pathogens are predominantly considered primary pathogenic, leading to immunosuppression, the release of pro-inflammatory cytokines or the destruction of the respiratory epithelium [4,5,6]. Thus, they facilitate the invasion of secondary pathogens, which are part of the physiological flora in the respiratory system [3]. Potential secondary pathogens include *Pasteurella* (*P*.) *multocida*, *Bordetella* (*B*.) *bronchiseptica*, *Glaesserella* (*G.*) *parasuis*, *M*. *hyorhinis* and *Streptococcus* (*S*.) *suis* [7,8]. However, the impact of different infectious agents can vary across different regions [1,9]. Non-infectious causatives, including a low ambient temperature, humidity, ventilation, dust, increased ammonia levels, overcrowding, mixing of pigs, increased stress, genetics, and probably other undescribed factors, also play a pivotal role in the pathogenesis of the PRDC but are usually difficult to measure [3].

On the other hand, only the detection of infectious agents via PCR or microbiological examination does not necessarily confirm disease, since pathogens like *A. pleuropneumoniae*, *G. parasuis*, or *M. hyorhinis* are also considered colonizers of the upper respiratory tract in healthy pigs [10,11,12,13,14,15]. In general, various specimens can be taken for confirmation of infections with pathogenic agents. For the detection of PRRSV numerous organs like lung, lymphatic tissue, tonsil, but also nasal swabs, oropharyngeal scrapings, oral fluids (OF), bronchoalveolar lavage fluids (BALF), processing fluids and serum can be used [16,17,18,19,20]. PCV2 is ubiquitous in swine stocks worldwide, and confirmation of disease depends on the occurrence of clinical signs, histologic lesions and the level of PCV2 DNA in lymphatic tissue [21]. Since thresholds for PCV2 DNA are only available for serum and samples of lymphoid tissues [22], proper interpretation of detected PCV2 DNA in BALF, OF, or swabs is not feasible [23,24]. The detection of IAV and *M. hyopneumoniae* is also described for lung tissue, tracheal transudate, BALF, OF, laryngeal swabs and nasal swabs [25,26]. While PCR is performed more frequently for the detection of *M. hyorhinis* in routine diagnostic procedures, isolation of *M. hyorhinis* is feasible and fast under certain conditions [15,27,28]. Since most *A. pleuropneumoniae* serotypes depend on nicotinamide adenine dinucleotide (NAD), growth requires either chocolate agar plates (CHOC) or a nurse streak [29]. Thus, besides isolation, PCR is widely applied for the detection of *A. pleuropneumoniae*. Similar to *A. pleuropneumoniae*, *G. parasuis* also depends on external NAD sources for growth [30]. However, especially the impact of *G. parasuis* and *M. hyorhinis* as potential causatives of respiratory disease is still controversial [13,14,31,32,33]. Since *P. multocida*, *B. bronchiseptica* and *S. suis* are frequently recovered from the respiratory tissues of healthy pigs, they are considered colonizers of the (upper) respiratory tract [34,35,36]. Thus, the isolation of either pathogen remains difficult to interpret. However, there is evidence that in case of concurrent infections, symptoms and lung lesions are more severe compared to mono-infections [6,34,37,38]. While *P. multocida* and *S. suis* grow easily on sheep-blood agar plates, *B. bronchiseptica* often gets overgrown by other bacteria, emphasizing the need for selective media for proper isolation [36]. As described, there is an abundance of specimens that can be taken for the detection of pathogens involved in the PRDC. However, some specimens could provide higher chances of pathogen detection. Thus, we aimed to evaluate how frequently certain pathogens were detected in different sample material in the course of routine diagnostic procedures within six years. In addition, we wanted to report the frequency of combinations of pathogens associated with the PRDC in samples from the same animal.

## 2. Materials and Methods

### 2.1. Study Design and Sample Collection

This is a retrospective study based on data from routine diagnostic samples from pigs with respiratory disease in Austrian swine stocks, which have been examined at the University of Veterinary Medicine in Vienna from January 2016 until December 2021. Within that time frame, 1975 samples were examined for different pathogens causing respiratory disease in pigs (Table 1). Altogether, 1010 samples were taken directly on the respective farms (n = 597) solely from pigs with respiratory symptoms by 82 different herd veterinarians and were sent to the University of Veterinary Medicine for diagnostic workup. The remaining 965 samples were taken in the facilities of the University of Veterinary Medicine from 514 pigs from 171 different Austrian swine stocks submitted for necropsy. From those animals, samples for diagnostic workup were taken immediately after euthanasia. The majority of all samples derived from swine stocks located in Upper Austria (n = 1012), while the remaining samples derived from swine stocks in Styria (n = 510), Lower Austria (n = 426), Carinthia (n = 19) and Burgenland (n = 8).

For the detection of PRRSV RNA, a pool consisting of lung tissue, tonsil and the tracheobronchial lymph node was taken exclusively from animals that were necropsied at the university. All nasal swabs, OF and BALF samples were submitted from the field.

Microbiological examination, cultivation for *M. hyorhinis*, as well as PCR for nucleotide detection of PRRSV, IAV, PCV2, *M. hyopneumoniae*, *M. hyorhinis*, *A. pleuropneumoniae* and *G. parasuis* were only performed if requested by the respective herd veterinarian (Table 1). Therefore, since analysed data derived from routine diagnostic samples, not all samples were tested for all selected pathogens. Altogether, only 23 lungs samples from 13 different farms were tested simultaneously for all pathogens. In this study, we only included samples from the respiratory tract and lymphatic system. Since infections with PCV2 or PRRSV are not limited to the respiratory tract, we aimed to include their detection in lymphatic tissue, as both pathogens are facilitators for infections with other pathogens involved in the PRDC [39]. Serum samples for the detection of PRRSV and PCV2 were not included in the study. Since there are currently no data on the overall prevalence of respiratory pathogens in Austrian swine stocks, we included all pathogens that were recovered from pigs with respiratory disease in routine diagnostic procedures. On the other hand, we excluded other pathogens like PRV, PCMV, CSFV and PRCV because they either do not occur in Austria (CSFV, PRV) or play a minor role in the PRDC (PCMV, PRCV). In addition to PCR, 72 lymph nodes and 15 lungs were also tested for PCV2 DNA by ISH.

### 2.2. Virological Investigations

For nucleic acid extraction, 100 mg of tissue sample was homogenized in 1 mL of PBS and four sterile 3 mm stainless steel beads in a Tissue Lyser II (Qiagen, Hilden, Germany). In the case of pools, equal parts of each tissue were included in the preparation. After centrifugation at 16,000× *g* for one minute, 140 µL of supernatant was extracted employing the QIAamp Viral RNA Mini QIAcube Kit in a QIAcube (Qiagen, Hilden, Germany). Nasal swabs were vortexed for 10 s in 1 mL of sterile PBS and 140 µL of swab suspension was used for extraction. OF and BALF samples were centrifuged at 16,000× *g* and 140 µL of supernatant was extracted. For RNA as well as DNA extraction, the same extraction kit was applied. Real-time PCRs were carried out in a Rotor-Gene Q 5-plex machine (Qiagen, Hilden, Germany) using primer and probe sequences and conditions shown in Table 2. Plasmid standards as well as blanks consisting of sample-free extracts, which were produced simultaneously to each extraction process, were tested side by side with the samples. A beta-actin mRNA RT-qPCR was performed for each sample extract to exclude PCR-inhibiting substances (Table 2).

### 2.3. Microbiological Investigations

For the cultural isolation of *M. hyorhinis*, specimens such as nasal swabs, 100 µL BALF, or 0.5 cm^3^ lung samples were placed into 900 µL 2SP medium, vortexed, and 100 µL of suspension were plated onto Friis and modified SP4 agar and incubated at 37 °C in a 5% CO_2_ atmosphere for up to 10 days [44]. Once cultivated, single mycoplasma colonies were picked, enriched in corresponding broth medium and identified using MALDI-TOF as described previously [45]. PCRs for the detection of *M. hyopneumoniae* and *M. hyorhinis* were performed as described previously [46,47]. PCR assays for the detection of DNA from *G. parasuis* and *apxIV* of *A. pleuropneumoniae* were also performed as described before [48,49].

For microbiological examination, all samples were incubated on Columbia agar III with 5% sheep blood (BA) (Becton Dickinson, Heidelberg, Germany) under aerobic and anaerobic conditions, BD CHOC under microaerobic conditions at 37 °C for 24–48 h (sometimes up to 72 h), CN (Colistin-Nalidixic Acid) agar with 5% sheep blood, improved II (Becton Dickinson), MacConkey II (MC) (Becton Dickinson) at 37 °C for 24–48 h, and on Sabouraud dextrose agar with gentamicin and chloramphenicol (SAB (Becton Dickinson)) at 28 °C for five days.

Samples were also incubated in a thioglycollate medium enriched with vitamin K1 and haemin (Becton Dickinson). All isolates were identified to the species level by matrix-assisted laser desorption-ionization-time of flight mass spectrometry (MALDI-TOF MS) (Bruker Daltonik, Bremen, Germany).

### 2.4. Evaluation and Statistical Analyses

We obtained all results retrospectively by using TIS^®^ (Tierspital-Informationssystem Orbis VetWare, Agfa HealthCare, Bonn, Germany). We included all 1975 samples deriving from pigs with respiratory problems from Austrian swine stocks being investigated from January 2016 to December 2021 at the University of Veterinary Medicine in Vienna. The test results of each sample were listed in Microsoft Excel^®^ (Appendix A). In the case of necropsied animals (n = 514), different samples of the same animal were grouped together in Excel. In case of submitted samples (n = 1010), evaluation was based on a sample level since information on individual animals was not provided. If a certain bacterium (*A. pleuropneumoniae*, *G. parasuis*, *P. multocida*, *B. bronchiseptica* and *S. suis*) could not be recovered from the respiratory sample on any agar plate, its examination result was defined as negative. For *M. hyorhinis*, *A. pleuropneumoniae* and *G. parasuis*, we obtained results from microbiological examination as well as PCR. Thus, for statistical tests, results for the detection of *M. hyorhinis*, *A. pleuropneumoniae* and *G. parasuis* in culture and PCR were combined. If a sample was investigated via microbiological examination and PCR for a certain pathogen, but the pathogen was detected only by one method, the result was categorized as positive. Positivity rates of the respective pathogens were obtained by dividing the number of specimens that were tested positively for the respective pathogen by the total number of specimens that were tested for the pathogen. For calculation of positivity rates and associations of concurrent infections, Pearson’s chi squared tests were assessed using SPSS^®^ (SPSS Statistics 25) [50]. Since not all samples were tested for each pathogen, potential associations of concurrent infections were only calculated for samples which were investigated for both respective pathogens.

## 3. Results

### 3.1. Positivity Rates

In absolute numbers, PRRSV was detected most frequently as it was positive in 271 of all 921 investigated samples, whereas *M. hyorhinis* had the highest positivity rate (=number of positive samples/number of tested samples) (55.1%), followed by PCV2 (38.0%) and *S. suis* (30.6%) (Figure 1). The positivity rate of *P. multocida* was more than twice as high as the positivity rate of *B. bronchiseptica.* Altogether, IAV-RNA was detected 29 times and had the lowest positivity rate (6.1%).

### 3.2. Positivity Rates of Pathogens in Different Specimens

PRRSV RNA was detected more frequently in tissue pools consisting of lung, tonsil and the tracheobronchial lymph node (36.2%) compared to other specimens such as lung tissue, OF and BALF (Table 3). Altogether, 30.6% of all investigated lung samples (185/605) were positive for *M. hyopneumoniae* DNA, while 3.4% of all investigated OF samples were positive for *M. hyopneumoniae* DNA (3/87). In contrast to *M. hyopneumoniae*, the positivity rates for *M. hyorhinis* and *S. suis* were higher in nasal swabs than in lung tissue. *G. parasuis* was never recovered from nasal swabs.

Out of all 197 samples which were positive for PCV2 DNA by quantitative (q) PCR, 65 samples had a viral load beyond 10^7^ genome equivalents per mg of tissue, which is considered the threshold for the confirmation of PCV2 systemic disease [22]. In addition to qPCR, 20/72 lymph nodes and 8/15 lung samples were also positive for PCV2 DNA by ISH. All samples which were tested positive by ISH were also positive by qPCR. PCR for the detection of *M. hyorhinis* DNA as well as microbiological examination on specific agars for the isolation of *M. hyorhinis* were both performed on 31 lungs. *M. hyorhinis* could be isolated from all lung tissue samples which were positive for *M. hyorhinis* DNA by PCR (n = 11). In addition, *M. hyorhinis* could also be recovered from 13 out of 20 lung samples, which were negative by PCR. In total, 111 lung tissue samples were tested via microbiological examination and PCR for the detection of *A. pleuropneumoniae* DNA. Out of those samples, all lungs positive for *A. pleuropneumoniae* in microbiological examination were also positive by PCR (n = 16). Ten out of 95 samples, from which *A. pleuropneumoniae* could not be isolated, were positive by PCR. While *G. parasuis* DNA was detected by PCR in 36 samples, it could be isolated via microbiological investigation from 24 samples. Out of those, *G. parasuis* was significantly more likely to be isolated from samples that were taken and examined immediately after euthanasia of pigs (n = 15/224) than from submitted samples from the field (n = 9/395) (*p* = 0.006) (Table 4).

### 3.3. Concurrent Infections

Samples being tested positive for PRRSV RNA were also significantly more likely to be positive for PCV2 (*p* = 0.001), *M. hyopneumoniae* (*p* = 0.032) and *P. multocida* (*p* < 0.001) (Table 5). *P. multocida* (*p <* 0.001) and *S. suis* (*p =* 0.46) were also recovered more frequently in samples that were positive for *M. hyopneumoniae*. At the same time, *M. hyopneumoniae*-positive samples were less frequently positive for *M. hyorhinis* (*p* = 0.004). An association of positively tested samples was also observed for *M. hyorhinis* and *S. suis* (*p* < 0.001). *P. multocida* (*p* = 0.048), *B. bronchiseptica* (*p* = 0.019) and *S. suis* (*p* < 0.001) were less likely to be recovered in *A. pleuropneumoniae*-positive samples.

## 4. Discussion

Since this is a retrospective study based on samples submitted for routine diagnostic procedures, there are multiple weaknesses limiting the overall evaluation of the obtained data. Firstly, there is a lack of information on the clinical history of pigs, from which samples derive. For instance, it was not possible to distinguish between samples from acutely diseased pigs and samples from chronically diseased animals. Overall, evaluating solely results from acutely diseased animals could have resulted in higher detection rates for the evaluated pathogens. In addition, information on age, actual symptoms, morbidity, mortality and vaccinations were not provided. Furthermore, we did not receive sufficient data on the antibiotic treatment of pigs prior to sample collection, aggravating especially the interpretation of results obtained from microbiological examinations. Since pathohistologic examination was not performed on all lung samples, the results of pathohistologic examination were excluded from this analysis. Thus, the interpretation of whether the detected pathogens had really caused respiratory symptoms or not was not feasible. In general, the comparability of detection rates among different specimens is very limited since specimens derived from different animals and different swine stocks. In particular, statements on detection rates of pathogens in BALF and nasal swabs should be done with caution due to the low sample size. In addition, it is possible that farms with a high number of submitted samples are overrepresented in the study. Despite the fact that serum may be taken most frequently for the detection of PRRSV RNA or PCV2 DNA, we decided to exclude all serum samples taken for the detection of PRRSV or PCV2, as both pathogens are also associated with other problems, including reproductive disorders, and we could not distinguish retrospectively between serum samples taken from pigs with respiratory disease from serum samples taken from pigs with other or no clinical signs. In addition, serum is frequently taken for the detection of PRRSV RNA for screening purposes without the presence of respiratory symptoms in the respective herd.

Furthermore, since several Austrian laboratories offer detection of pathogens occurring in swine stocks, certain herd veterinarians may prefer other laboratories and do not submit samples to our facilities at all. The choice of the laboratory may also depend on funding, which varies among different federal states in Austria. Therefore, federal states supporting funding of diagnostics may also be overrepresented compared to federal states without funding. Indeed, Upper Austria is overrepresented in the current study, since 51% of all analysed respiratory samples derived from swine stocks in Upper Austria, whereas approximately 40% of all Austrian pigs are kept in Upper Austria. On the other hand, Lower Austria and Carinthia, where 28% and 4% of all pigs are kept, respectively, are underrepresented.

In addition, the storage, time and mode of transport of samples also have a substantial impact on the output of diagnostic approaches, in particular for microbiological investigations. Also, the DNA and RNA of most analysed pathogens degrade faster under warmer conditions. Furthermore, PCR for the detection of certain pathogens or microbiological examination was only performed if requested by the submitting herd veterinarian for routine diagnostic purposes. Since solely 23 lungs were tested for all pathogens that were analysed in the current study and a total of 106 samples were only tested for a single pathogen, the overall statement on the frequency of certain coinfections is very limited. In addition, submitted samples may be tested more frequently for pathogens that were considered to be more important by the respective herd veterinarian. In addition, multiple pairwise comparison tests were not performed. Retrospective investigations of samples for pathogens that were previously not tested in routine diagnostic procedures were not feasible since most of the samples were not archived. Thus, our data cannot provide an overview of the prevalence of certain pathogens in Austria. While our data definitely cannot provide an overview of the final diagnosis, it clearly shows the detection rates of certain pathogens in diseased animals. Knowledge on the occurrence of pathogens associated with the PRDC in animals with respiratory symptoms is pivotal for veterinarians.

The fact that PRRSV was detected most frequently could be attributed to the fact that PRRSV is one of our major research topics. Therefore, our facilities may provide a better diagnostic workup for PRRSV compared to IAV due to our experience working with PRRSV. Nonetheless, our data demonstrate that PRRSV RNA was detected most frequently in tissue samples consisting of lung and lymphatic tissue. Thus, adding lymphatic tissue can increase the odds of detecting PRRSV RNA, as PRRSV is predominantly replicating in lymphatic tissue during its stage of persistence [51,52]. However, since all pools consisting of lung and lymphatic tissues were taken directly from the facilities of the university, a higher detection rate for PRRSV in the pool could also be due to testing for PRRSV immediately after euthanasia of the respective animal without shipment. We assume that low positivity rates of PRRSV in OF are likely due to dilution effects and/or inhibitory substances contained in saliva. In addition, sample preparation and RNA extraction methods also ought to be adapted for OF [53]. Furthermore, most OF, which derive from 67 different farms, were collected for screening purposes and not for the diagnostic workup of an acute outbreak of respiratory disease. Noteworthy, the fact that all OF and BALF derived from submitted samples from the field, whereas all pool samples consisting of lung and lymphatic tissue were taken from pigs sampled directly at the university, may also explain the observed variation in the detection rate, as more RNA may have been degraded in submitted samples during transportation.

In contrast to PRRSV, the positivity rates of IAV RNA were similar in lung tissue, OF and nasal swabs. However, comparability of IAV detection rates amongst different specimens is limited, as IAV RNA was only detected in a total of 29 samples. Low positivity rates for IAV RNA (6.1%) from all samples that were tested for the pathogen (n = 479) were already demonstrated before [34,54]. This could be linked to various reasons. Firstly, direct detection of IAV is only feasible within the first four days post-infection due to its short duration of shedding [55,56]. Thus, the shedding of IAV might have already ceased in the sampled animals, which expressed respiratory symptoms probably due to secondary infections at the time of sampling. In addition, none of the submitted swabs were transferred into tubes with transport medium after sampling [57]. Differences in positivity rates for PCV2 DNA between lung and lymphatic tissue were not observed before [58]. However, samples were considered PCV2 positive based on the general detection of PCV2-DNA but not on the quantity of detected PCV2 DNA.

High positivity rates for PCV2 and *M. hyopneumoniae* can be explained by the fact that those pathogens are ubiquitous in swine stocks worldwide [59,60,61]. Our observations for positivity rates of *M. hyopneumoniae* DNA in different specimens go in line with previous findings, also describing that *M. hyopneumoniae* DNA was detected less frequently in OF compared to samples from the lower respiratory tract [26,62]. Our data also provide evidence that the isolation of *M. hyorhinis* from respiratory tissue is more sensitive than PCR, going in line with a previous study conducted in Thailand [28]. The high positivity rates in *M. hyorhinis* may be attributed to the fact that *M. hyorhinis* is a colonizer of the respiratory tract. High variation of positivity rates for *A. pleuropneumoniae* in pigs with respiratory symptoms amongst different studies (0.8% to 20.7%) could be explained by different sampling and different diagnostic approaches [34,63]. Although *G. parasuis* is a colonizer of the respiratory tract, we could never isolate it from nasal swabs [64]. However, a PCR for the detection of *G. parasuis* DNA from nasal swabs was never performed, and *G. parasuis* was probably overgrown by other bacteria occurring in the nasal cavities during microbiological examinations [65]. The more frequent isolation of *G. parasuis* from samples that were taken from pigs immediately after euthanasia at the University of Veterinary Medicine compared to submitted samples from the field is very likely due to the limited survival of *G. parasuis* outside the host [30,36]. This emphasizes that euthanasia and immediate sampling of acutely diseased animals shall be preferred over sampling of carcasses. In contrast to our observations, the recovery rates of *P. multocida* and *B. bronchiseptica* in lung tissue were relatively even in similar investigations [34]. Lower recovery rates of *B. bronchiseptica* may be a result of overgrowth by other faster-growing bacteria [66]. Furthermore, most submitted samples derived from fattening pigs, and *P. multocida* is usually more abundant in the upper respiratory tract of fattening pigs compared to *B. bronchiseptica* [39,63].

Albeit confirming the frequent occurrence of multiple coinfections, it is pivotal to emphasize that confirmation of coinfections does not necessarily imply interactions among those pathogens. In particular, the role of certain colonizers in the pathogenesis of the PRDC ought to be investigated in the future. Similar to our results, an American study also reported that PRRSV RNA was detected in 51.9% of all piglets that were PCV2-positive [67]. Our observations on frequent coinfections involving PRRSV and *M. hyopneumoniae* coincide with several other investigations [1,34,68,69]. In previous studies, *M. hyopneumoniae* and *P. multocida* were also detected more frequently in PRRSV-positive lungs [34,69].

In accordance with our results, investigations of lungs from slaughtered pigs in France also reported that *P. multocida* was more likely to be detected in *M. hyopneumoniae*-positive lungs [63]. Especially, *M. hyopneumoniae* and *P. multocida* coinfections are frequently described in pigs with respiratory disease [34,37,63]. In contrast to our findings, investigations of lungs from slaughter pigs revealed no correlation between detection of *M. hyopneumoniae* and *M. hyorhinis*, while another study examining BALF samples from pigs with respiratory problems reported a positive association [34,70]. Similar to our results, a positive association between *M. hyorhinis* and *S. suis* has also been described before [34]. While it is possible that *M. hyorhinis* could act as a precursor for secondary pathogenic agents in a similar way as *M. hyopneumoniae*, its potential impact on the PRDC has not been fully elucidated yet [32]. Since the impact of *G. parasuis* on the respiratory disease of pigs is also controversial and widely discussed, it was less surprising that *G. parasuis* was not detected frequently in the same samples as other pathogens [30,34]. In general, coinfections with *P. multocida* and *B. bronchiseptica* were less frequent than expected, which could be due to the fact that *B. bronchiseptica* gets overgrown easily in the course of cultivation [66]. Furthermore, calculated negative associations among bacteria are probably a consequence of data collection, as the examination results of certain bacteria were defined as negative if they could not be recovered from any agar plate. In addition, *A. pleuropneumoniae*, *G. parasuis* and *B. bronchiseptica* are more difficult to cultivate due to special growth requirements and easier overgrowth by other bacteria compared to *P. multocida* and *S. suis*, which were recovered together frequently.

In conclusion, positivity rates varied among the different sample materials. In addition, lung samples that were positive for a primary pathogenic agent like PRRSV or *M. hyopneumoniae* were more likely to be positive for a secondary pathogenic agent like *P. multocida* or *S. suis*.

## Figures and Tables

**Figure 1 vetsci-10-00601-f001:**
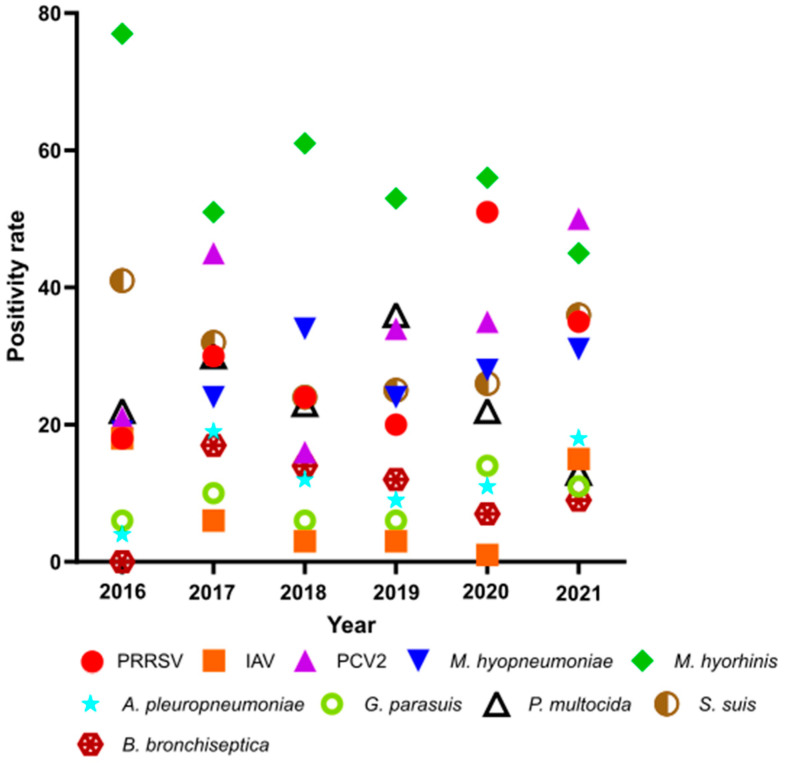
Positivity rates of each pathogenic agent from 2016 to 2021. PRRSV = porcine reproductive and respiratory syndrome virus, IAV = influenza A virus, PCV2 = porcine circovirus type 2, *M. hyopneumoniae* = *Mycoplasma hyopneumoniae*, *M. hyorhinis* = *Mycoplasma hyorhinis*, *A. pleuropneumoniae* = *Actinobacillus pleuropneumoniae*, *G. parasuis* = *Glaesserella parasuis*, *P. multocida* = *Pasteurella multocida*, *S. suis* = *Streptococcus suis*, *B. bronchiseptica* = *Bordetella bronchiseptica*.

**Table 1 vetsci-10-00601-t001:** Number of samples that were tested by the respective diagnostic method.

Investigations	Lung	Lymph Node	Lung, Tonsil, Lymph Node	BALF	OF	Nasal Swabs	Total
Microbiological examination	590			21		23	634
*M. hyorhinis* cultivation	142			17		8	167
PRRSV-PCR	278		469	38	136		921
IAV-PCR	240		51	12	94	82	479
PCV2-PCR	168	329	9	10	2	0	518
*M. hyopneumoniae*-PCR	605			10	87	11	713
*M. hyorhinis*-PCR	139			36		5	180
*A. pleuropneumoniae*-PCR	193			3		2	198
*G. parasuis*-PCR	157			8			165
Samples (Total)	874	329	469	56	142	105	1975

BALF = bronchoalveolar lavage fluids, OF = oral fluids, *M. hyorhinis* = *Mycoplasma hyorhinis*, PRRSV = porcine reproductive and respiratory syndrome virus, IAV = influenza A virus, PCV2 = porcine circovirus 2, *M. hyopneumoniae* = *Mycoplasma hyopneumoniae*, *A. pleuropneumoniae* = *Actinobacillus pleuropneumoniae*, *G. parasuis* = *Glaesserella parasuis*.

**Table 2 vetsci-10-00601-t002:** PCR kits and primers applied for detection of viruses associated with the PRDC.

Virus	Primer/Probe Sequences	PCR Kit	Reference
PRRSV	Primer-F (PRS133): 5′-ATGGCCAGCCAGTCAATC-3′	One Taq One-Step RT-PCR Kit (NEB)	[40]
	Primer-R (PRS134): 5′-TCGCCCTAATTGAATAGGTG-3′		
IAV	FLUAM-1F: 5′-AAGACCAATCCTGTCACCTCTGA-3′	Luna^®^ Universal Probe One-Step RT-qPCR Kit (NEB)	[41]
	FLUAM-2F: 5′-CATTGGGATCTTGCACTTGATATT-3′		
	FLUAM-1R: 5′-CAA AGCGTCTACGCTGCAGTCC-3′		
	FLUAM-2R: 5′-AAACCGTATTTAAGGCGACGATAA-3′		
	FLUAM-1P: FAM-5′-TTTGTGTTCACGCTCACCGT-3′-TAMRA		
	FLUAM-2P: FAM-5′-TGGATTCTTGATCGTCTTTTCTTCAAATGCA-3′-TAMRA		
PCV2	qPCV2-F: 5′-GAGTCTGGTGACCGTTGCA-3′	Luna^®^ Universal Probe One-Step RT-qPCR Kit (NEB)	[42]
	qPCV2-R: 5′-YCCCGCTCACTTTCAAAAGTTC-3′		
	qPCV2-probe: FAM-5′-CCCTGTAACGTTTGTCAGAAATTTCCGCG-3′-BHQ1		
ß-Actin	qBeta-Actin-1005-F: 5′-CAGCACAATGAAGATCAAGATCATC-3′	Luna^®^ Universal Probe One-Step RT-qPCR Kit (NEB)	[43]
	qBeta-Actin-1135-R: 5′-CGGACTCATCGTACTCCTGCTT-3′		
	qBeta-Actin-1081-probe: HEX-5′-TCGCTGTCCACCTTCCAGCAGATGT-3′-BHQ1		

PRRSV = porcine reproductive and respiratory syndrome virus, IAV = influenza A virus, PCV2 = porcine circovirus 2.

**Table 3 vetsci-10-00601-t003:** Number and percentage of positively tested samples via PCR or microbiological examination in different kinds of specimens and total number of tested samples for each pathogen.

	Lung	Lymph Node	Lung, Tonsil, Lymph Node	BALF	OF	Nasal Swabs	Total
PRRSV	92/278		170/469	5/38	4/136		271/921
	33.1%		36.2%	13.2%	2.9%		29.4%
IAV	18/240		4/51	0/12	2/94	5/82	29/479
	7.5%		7.8%	0%	2.1%	6.1%	6.1%
PCV2	79/168	111/329	7/9	0/10	0/2		197/518
	47%	33.7%	77.8%	0%	0%		38.0%
*M. hyopneumoniae*	185/605			0/10	3/87	0/11	188/713
	30.6%			0%	3.4%	0%	26.6%
*M. hyorhinis*	136/257			26/46		12/13	174/316
	52.9%			56.6%		92.3%	55.1%
*A. pleuropneumoniae*	83/679			0/21		3/23	86/723
	12.2%			0%		13%	11.9%
*G. parasuis*	52/641			8/23		0/23	60/
	8.1%			34.8%		0%	8.8%
*P. multocida*	146/590			0/21		5/23	151/634
	24.7%			0%		21.7%	23.8%
*B. bronchiseptica*	62/590			3/21		1/23	66/634
	10.5%			14.3%		4.3%	10.4%
*S. suis*	163/590			18/21		13/23	194/634
	27.6%			85.7%		56.5%	30.6%

BALF = bronchoalveolar lavage fluids, OF = oral fluids, PRRSV = porcine reproductive and respiratory syndrome virus, IAV = influenza A virus, PCV2 = porcine circovirus type 2, *M. hyopneumoniae* = *Mycoplasma hyopneumoniae*, *M. hyorhinis* = *Mycoplasma hyorhinis*, *A. pleuropneumoniae* = *Actinobacillus pleuropneumoniae*, *G. parasuis* = *Glaesserella parasuis, P. multocida* = *Pasteurella multocida*, *B. bronchiseptica* = *Bordetella bronchiseptica*, *S. suis* = *Streptococcus suis.*

**Table 4 vetsci-10-00601-t004:** Results of investigations for *Glaesserella parasuis*.

Investigations	Result	Field Samples	Samples from Animals Submitted for Necropsy
Isolation	Negative	386	209
	Positive	9	15
PCR	Negative	66	64
	Positive	27	12

**Table 5 vetsci-10-00601-t005:** Combinations of detected pathogens out of all investigated samples.

	Number of Samples Being Positively Tested for Both Pathogens	Number of Samples Being Tested for Both Pathogens	Positivity Rate	*p*-Value
*P. multocida* and *S. suis*	151	634	24%	0.517
PRRSV and PCV2	66	368	18%	**0.001**
*M. hyorhinis* and *S. suis*	66	193	34%	**<0.001**
PRRSV and *P. multocida*	52	347	15%	**<0.001**
*M. hyopneumoniae* and *P. multocida*	41	330	12%	**<0.001**
PRRSV and *M. hyopneumoniae*	41	426	10%	**0.032**
PRRSV and *S. suis*	40	347	12%	0.388
*M. hyopneumoniae* and *S. suis*	37	330	11%	**0.046**
PRRSV and *A. pleuropneumoniae*	31	391	8%	**0.001**
PCV2 and *M. hyorhinis*	29	125	23%	0.256
PCV2 and *S. suis*	29	242	12%	0.398
PRRSV and *M. hyorhinis*	26	191	14%	0.092
*M. hyorhinis* and *P. multocida*	25	193	13%	0.378
*M. hyorhinis* and *G. parasuis*	24	212	11%	0.509
PCV2 and *P. multocida*	24	242	10%	0.856
PCV2 and *M. hyopneumoniae*	24	265	9%	0.397
*M. hyopneumoniae* and *M. hyorhinis*	23	218	11%	**0.004**
PCV2 and *A. pleuropneumoniae*	21	272	8%	**0.002**
*G. parasuis* and *S. suis*	18	634	3%	0.793
PRRSV and *G. parasuis*	15	373	4%	0.562
*A. pleuropneumoniae* and *S. suis*	14	634	2%	**0.017**
PRRSV and *B. bronchiseptica*	12	347	3%	0.667
PCV2 and *B. bronchiseptica*	11	242	5%	0.412
*M. hyopneumoniae* and *B. bronchiseptica*	11	330	3%	0.464
*M. hyorhinis* and *B. bronchiseptica*	11	193	6%	0.696
*A. pleuropneumoniae* and *P. multocida*	11	634	2%	**0.048**
*B. bronchiseptica* and *S. suis*	11	634	2%	**0.009**
PCV2 and *G. parasuis*	9	264	3%	0.144
PRRSV and IAV	8	364	2%	0.727
IAV and *S. suis*	8	185	4%	0.724
*M. hyopneumoniae* and *G. parasuis*	8	358	4%	0.273
*G. parasuis* and *P. multocida*	8	634	1%	0.079
IAV and *M. hyorhinis*	7	109	6%	0.861
IAV and PCV2	5	149	3%	0.811
IAV and *P. multocida*	4	185	2%	0.502
*P. multocida* and *B. bronchiseptica*	4	634	1%	**<0.001**
IAV and *M. hyopneumoniae*	3	291	1%	0.681
IAV and *G. parasuis*	2	194	2%	0.704
IAV and *A. pleuropneumoniae*	2	213	1%	0.800
*M. hyopneumoniae* and *A. pleuropneumoniae*	2	398	1%	**<0.001**
*M. hyorhinis* and *A. pleuropneumoniae*	2	209	1%	**0.038**
*A. pleuropneumoniae* and *B. bronchiseptica*	2	634	0%	**0.019**
*G. parasuis* and *B. bronchiseptica*	2	634	0%	0.079
*A. pleuropneumoniae* and *G. parasuis*	1	649	0%	**0.014**

*P. multocida* = *Pasteurella multocida*, *S. suis* = *Streptococcus suis*, PRRSV = porcine reproductive and respiratory syndrome virus, PCV2 = porcine circovirus type 2, *M. hyorhinis* = *Mycoplasma hyorhinis*, *M. hyopneumoniae* = *Mycoplasma hyopneumoniae*, *G. parasuis* = *Glaesserella parasuis*, *B. bronchiseptica* = *Bordetella bronchiseptica*, IAV = influenza A virus, bold: significant associations (*p* < 0.05), red: positive association, blue: negative association.

## Data Availability

All collected data are available as Appendix A.

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
