# Peer review of "Retrospective Analysis of the Detection of Pathogens Associated with the Porcine Respiratory Disease Complex in Routine Diagnostic Samples from Austrian Swine Stocks"

_vetsci, 2023, doi:10.3390/vetsci10100601_

Round 1

Reviewer 1 Report

In the manuscript entitled "Retrospective analysis of the detection of pathogens associated with the porcine respiratory disease complex in routine diagnostic samples", the authors aim to compare the detection rates of certain viruses and bacteria in routine diagnostic samples from pigs with respiratory problems. Here are concerns as outlined for the authors to consider below.

Major issues

  1.  The study design of this manuscript seems not very reasonable. The sample populations for detection of individual pathogens are different. For example, among 874 lung samples, 278 samples are tested for PRRSV, 240 samples are tested for IAV, 168 samples are tested for PCV, ….  How many lung samples are tested simultaneously for all selected pathogens? If the authors want to compare positive rates of different pathogens in lung samples of pigs, it is better to test all these pathogens in all individual lung samples. Otherwise, the detection rates of pathogens in different sample populations are not comparable.

  2.  The description of results in this manuscript is not rigorous.

a.       The authors state that “From 2016 to 2021, PRRSV was detected most frequently, and it was positive in 29.4% of all investigated samples (Table 3).” Actually, the M. hyorhinis has the highest positivity rate (55.1%). Why do the authors say that PRRSV is detected most frequently?

b.       Similarly, the authors state that “Tissue pools consisting of lung, tonsil and the tracheobronchial lymph node had the highest positivity rate for the detection of PRRSV RNA (36.2%) (Table 3).” However, the highest positive rate is observed in the detection of PCV2 RNA (77.8%) (Table 3).

c.       The authors state that “PRRSV RNA was detected more frequently in tissue samples than in BALF or OF samples”. Since the tissue sample and the corresponding BALF or OF sample are not collected from one same pig, the positive rates of PRRSV RNA in different kinds of specimens may be not comparable.

  3.  The authors also analyze the associations between different pathogens in pigs. Similarly, the co-infection rates of the pathogens are not comparable because the tested sample populations are different.

  4.   The authors conclude that “In conclusion, detection rates varied among pathogens and specimens, suggesting diverse explanations for observed differences”. Please provide a specific conclusion for this study.

No comments

Author Response

Dear reviewer, 

the authors thank you for the review report. 

All responses are attached in the file below. 

Reviewer 2 Report

This study aimed to 1) compare the positivity rates of pathogens involved in the PRDC in different routine diagnostic specimens and 2) to reflect the current situation of coinfections with respiratory pathogens in the field. Regarding the first goal, it seems the authors mean to estimate positivity rate in different specimens to ascertain which sample types yield best probability of detecting diseases. However, if this is the case, a database of submitted samples can underestimate the probability of detecting the pathogen on a given sample type since the animals might not actually be infected (samples could be truly negative). If that is the goal, different sample types from known positive animals should have been used. Additionally, the course of infection when the sample was taken will likely play a role no probability of detection in the different sample types, information which seemed to no have been available to the authors, as mentioned in the discussion. Regarding the second goal, authors imply by how the goal was phrased that this is an estimation of coinfections prevalence in the field. However, as mentioned in the discussion, veterinarian might opt for submitting samples for diagnosis elsewhere. The authors provide no clarification as to what population this data represents (is it representative of a given region? How much coverage does this particular lab has regarding swine production in Austria?).

Additional comments:

- About half the samples were collected in the field, the other half was collected in the University facilities. What does that mean? Were they collected from University maintained herds? Were they collected from research herds from experimental studies? Were they collected from field animals sent for necropsy at the Veterinary Diagnostic Lab? The characterization of this population is crucial to understand the underlying probability of these animals actually testing positive for any of these diseases.

- Some important limitations on the coinfection analysis were not mentioned. Microbiological testing was only done if requested by the vet, meaning not all samples had the same probability of detecting coinfections (vets might be more prone to request testing if there is a strong clinical indication of infection, meaning probability of testing positive is likely higher from the beginning). Authors excluded PRRS and PCV2 serum results. I don’t understand why since it is likely one of the most common sample types but this potentially reduced they capability of detecting PRRS or PCV2 coinfections.

- Authors report that tissue pools had higher positivity rate for PRRS but this specimen was only collected from animals that were sampled at the University. We need a better description on what that population represents to be able to interpret this result. This also applies to the description of G. parasuis more likely to be isolated after euthanasia than from field samples.

- Concurrent infections were made by a series of two-by-two comparisons. The more inferences are made, the more likely to find spurious correlations. Authors should consider using multiple pairwise comparison tests

- Two of their conclusions seem overstated based on the issues I have presented when addressing each of the goals. “Consequently, our study provides important information for practitioners for the choice of proper specimens” and “Despite having a plethora of limitations, the evaluated data demonstrate the odds to detect certain pathogens depending on the sampling material under field conditions”

-Authors state that “In general, coinfections with P. multocida and B. bronchiseptica were less frequent than expected. This could be due the fact that fattening pigs are more likely to be infected with P. multocida, while B. bronchiseptica is more frequently recovered from piglets” on their discussion. This last phrase does not make sense since they had no information on age of the animals from which the samples came from, according to their first discussion paragraph.

Author Response

The authors thank the reviewer for submitting their reviewer report. 

Responses are provided in the attached file "Reviewer 2" below.

Reviewer 3 Report

The authors did a retrospective (2016-2021) analysis of the detection of pathogens associated with the porcine respiratory disease complex in routine diagnostic samples. This is a very interesting study although as indicated by authors had several limitations. A future study might be initiated based on the results to look into co-infection with various diseases in porcine with respiratory symptoms.

I have a few suggestions/questions on the paper. See below:

Simple summary, line 21: Suggest to delete “36.2% positive”.

Materials and methods, section 2.2, line 141-142: You describe in line 138 “the QIAamp Viral RNA Mini QIAcube Kit in a QIAcube (Qiagen, Hilden, Germany)” was used for extraction. Do you have any references where this has been used for both RNA and DNA extraction as described in your study? How do you know you did get DNA during extraction?

Materials and methods, section 2.2, line 144-145: You used plasmid standards as controls in the study. Did you include another sample Positive control which went through the extraction process similar to other samples? This would indicate if your sample extractions did indeed work for both RNA and DNA.

Table 2: You do not have a probe for PRRSV, so did you do conventional PCR for detection or can you add the probe?

I would include a concluding paragraph at the end of Discussion section.

Spelling and word use should be checked

Author Response

The authors thank the reviewer for submitting their reviewer report. 

Responses are provided in the attached file below.

Reviewer 4 Report

The submitted paper, Retrospective Analysis of the Detection of Pathogens associated with the Porcine Respiratory Disease Complex in Routine Diagnostic Samples, can contribute to the knowledge on detection trends of main pathogens of PRDC. Based on the shown data, the paper describes detection rates by specific sample types collected from pigs from swine stock. The main concern is the lack of clarity on the paper's goal and main conclusions (derived from the data analyzed). I would accept major revisions on population and data description, statistical analyses, and possible conclusions. 

Simple summary/abstract: it starts by stating about DIAGNOSING respiratory disease and then says, "detecting those viruses and bacteria can sometimes be quite difficult." The focus of the paper it seems porcine reproductive and respiratory syndrome virus (PRRSV) (n=921), influenza A virus (n=479), porcine circovirus type 2 (PCV2) (n=518), Mycoplasma (M.) hyopneumoniae (n=713), Actinobacillus pleuropneumoniae (n=198), Glaesserella (G.) parasuis (n=165) and M. hyorhinis (n=180) ... my question; Is difficult to detect PRRSV? At least in the USA, several PCRs can be used for various sample types (processing fluids, oral fluids, tongue tips, serum, family oral fluids, tonsils, etc) depending on the pig production phase and objective of sampling. I partially agree regarding IAV, given it is acute and short infection … less window time, but it's not challenging. You see IAV-like clinical signs you detect by PCR. Same thing with PCV2, as every pig has PCV2. On the bacteria, isolation depending on the bacteria is quite difficult – at least M hyo and MHR, but the isolation of GPS and APP is difficult. What is really difficult regarding the bacterial agents is to imply respiratory disease based on only the detection of specific sample types. The problem with the simple summary/abstract is that the paper starts talking about respiratory disease, but it is reporting detection. PRRSV, PCV2, GPS, APP, and MHR cause respiratory disease, and their main problem is systemic infection (polyserositis and sepsis by GPS, APP, MHR). The abstract's conclusion is pretty vague and does not say anything. What is the take-home of the paper? 

Introduction: clearer and states correctly that detection does not mean disease. Provide good background on the agents regarding PRSC and respiratory context …. But GPS, MHR, PRRSV, and PCV2 can cause systemic disease.

Objective: it isn't very clear because it is not aligned with the results in the simple summary/abstract. A simple summary/abstract shows DETECTION RATES and does not show any disease rate…. So how can the study's goal be "we aimed to reflect the current situation of coinfections with respiratory pathogens in the field"… now detection means disease? But the title is accurate with the results. Co-detection then? 

Materials and Methods: Were 1975 samples only respiratory? What is the percentage of respiratory cases compared to the rest? What type of cases do you diagnose in the lab? The paper needs to clarify the value of the findings by showing denominators. 

Line 124: "Other diagnostic methods like ISH were also excluded from the study due to low numbers of investigated samples." Why? The ISH would provide evidence if the case is a true respiratory problem or only detects the agent. Again … check the objective of the study. 

Line 104: add swine stock to the title… you need to state Austria's classification of swine stock and provide basic health characteristics (describe the study population). 

Line 183 says coinfection… how can you coinfection if you have a definitive disease diagnosis? If you are truly describing the disease, you must clarify how all the cases were diagnosed and which criteria were used to diagnose the final respiratory diagnoses. 

Line 192: Positive rate … do you have the information, such as Y was positive from X lungs for MHR isolation? 

Stats: To improve the work, calculate the positivity rate by year and show that in line plots for each agent 

Line 202: It is confusing … PRRSV was detected most frequently, and it was positive in 29.4%, but M. hyorhinis had the highest positivity rate (55.1%)? Can you clarify? Most tested? 

Discussion: it is well stated that the paper is regarding detection. It is overall well-written. 

Line 287: It says: "Consequently, our study provides important information for practitioners for the choice of proper specimens." This is true in diagnosing disease for some pathogens. What is the goal of the authors saying this? Would it not be to direct the practitioner to collect specific sample types to diagnose disease? I think you are over-concluding because you did not show any evidence/data of true respiratory cases to then say the practitioner to collect that sample to diagnose that specific agent. I will rephrase for specific agents that you might draw that conclusion. Example … MHR in nasal; 92.3% positivity rate … is this the best sample to diagnose MHR respiratory disease? S. suis only shows specimens where S. suis is commensal and may not be causing respiratory problems (lung, balf, and nasal). If you show which are the serotypes or sequence types with a history of causing disease, that may be plausible. 

Conclusions: where are the conclusions of the study? Can you add the main take homes in a new conclusion section or create a conclusion paragraph? 

Supplementary material: what is the goal of the Excel file? Any legend? What is the meaning of the numbers? Is this the raw data? 

Author Response

(The authors gave the same response as above.)

Reviewer 5 Report

All samples derived from pigs with respiratory problems and were investigated at the 18
University of Veterinary Medicine in Vienna between 2016 and 2021

Regarding this comment, it is not clear why samples were selected only from 2016-2021, if the paper is submitted in 2023, where are the sample data from 2022-2023

PCR, quantification data should be included.

Author Response

(The authors gave the same response as above.)

Round 2

Reviewer 1 Report

In this revised manuscript, there are still major issues that need to be addressed.

1, In the Simple Summary, the authors summarize that “RNA of the porcine reproductive and respiratory syndrome virus (PRRSV) was detected most frequently in tissue samples consisting of lung, tonsil and the tracheobronchial lymph node”. Though the authors have shown the results that the positive rate of PRRSV in the sample pool is the highest among different sample types, the results only imply that PRRSV RNA is detected more likely in the sample pool than other sample types. Since the sample number, origin, background, and storage for individual sample types are diverse, it is unconvincing, based on the data provided in this study, to show readers that there is higher probability of detection of PRRSV in the sample pool than in other sample types. To further confirm this hypothesis, the authors may analyze the data generated from different sample types of known individual positive pigs. The same issue is in the summary that “In total, most pathogens were detected more frequently in tissue samples compared to oral fluids or bronchoalveolar lavage fluids”.

2, In the Simple Summary, the authors summarize that “Glaesserella parasuis was isolated more frequently from samples taken immediately after euthanasia of the pig”. The authors describe the results in the main text, but there are no supporting data in the tables.

3, In the abstract, as described by the authors “Altogether, 1975 routine diagnostic samples from pigs in Austrian swine stocks between 2016 and 2021 were analyzed”, I do suggest the authors to further analyze the positivity rates of PRDC associated pathogens by year.

4, The conclusion in the abstract is neither proper nor convincing.

No

Author Response

The authors thank the reviewer for the valuable comments.

Responses to the revisions are provided in the attached document: "Revisions 2 _ Responses to reviewer 1"

All changes of the manuscript are marked in yellow colour. 

Reviewer 4 Report

The authors responded to the majority of comments. The comments that are not answered properly are:

Line 104: add swine stock to the title… You need to state Austria's classification of swine stock and provide basic health characteristics (describe the study population). 

I have asked to author as a brief description of the health characteristics of swine stock farms of Austria then the readers can be aware of the study population and apply the findings of the paper. The authors provided an extensive description that can summarized by targeting only swine stock HEALTH status characteristics  (PRRSV statuses? prevalence in stock farms? any free farms? Target information that is relevant to the work. 

Line 192: To improve the work, calculate the positivity rate by year and show that in line plots for each agent 

Can you select the main agent-sample type? Like, Austria uses a low number of oral fluids for PRRSV, Why? Can you show the trend? Select a few examples that are interesting of agent-sample type to show the trends by year.

I do not know if I missed that in the last revision, but the updated conclusion is not possible to be drawn.

"In conclusion, our data emphasize that taking tissue samples provides a higher probability to detect pathogens associated with respiratory disease in pigs compared to BALF or OF samples. In addition, testing for pathogens immediately after euthanasia of animals can help to increase the detection rates of certain pathogens such as G. parasuis."

How can you say higher probability of detection if you do not have the same sampling methodology for all sample types? Additionally, the study is pretty biased as you are using sample types submitted by vets in one lab. The only conclusion that can be drawn from the whole study is one sample type had a higher or lower frequency of positive results compared to the other one. That is the only conclusion possible. 

Author Response

The authors thank the reviewer for the valuable comments.

Responses to the revisions are provided in the attached document: "Revisions 2 _ Responses to reviewer 4"

All changes of the manuscript are marked in yellow colour. 
